# Validation of the Urban Walkability Perception Questionnaire (UWPQ) in the Balearic Islands

**DOI:** 10.3390/ijerph17186631

**Published:** 2020-09-11

**Authors:** Guillem Artigues, Sara Mateo, Maria Ramos, Elena Cabeza

**Affiliations:** Balearic Islands Public Health Department, Balearic Islands Public Health Research Group (GISPIB), Health Research Institute of the Balearic Islands (IdISBa), 07010 Palma, Spain; saramateoerroz@hotmail.com (S.M.); mramos@dgsanita.caib.es (M.R.); cabezaelena@gmail.com (E.C.)

**Keywords:** healthy routes, walkability, environment perception, health promotion, physical activity, neighborhood

## Abstract

Within the context of promoting the “healthy routes” program, the aim of this study was to validate the urban walkability perception questionnaire (UWPQ) in the Balearic Islands to determine the characteristics of the urban environment that promote walking among the population. The UWPQ measures pedestrian facilities, infrastructures of the environment, perception of safety and a participant’s general opinion. This process was performed in 12 routes predefined by a community participation program and set around the primary health centers. Degree of correlation between the items was calculated. The final internal consistency was 0.8 in all blocks according to the Cronbach’s alpha test (*p* < 0.01). Goodman and Kruskal–gamma correlation coefficient (γ) between the item measuring the general opinion and the rest of the items was significant. The items from the perception of safety and pedestrian facilities blocks were the ones that most affected the final assessment. Those regarding the pedestrian-only pavements, clearly marked pavements, noise, traffic density and parks condition obtained the lowest coefficients. To conclude, the results showed that the UWPQ is a suitable instrument to assess the degree of adequacy of the urban environment for walking. It could contribute to create healthy environments as well as to improve public policies.

## 1. Introduction

Walking is adults’ preferred and most practiced way of doing physical activity [1,2] that they would rather do close to their homes [3]. Walking regularly improves adults’ cardiovascular, metabolic, psychological and social health [4,5,6,7,8,9,10,11,12,13,14,15]. However, the prevalence of sedentarism is high and 35.2–38.8% of the world population does not do the required amount of physical activity needed to be in good health [16]. In our environment, 40% of the adult population living in the Balearic Islands claimed to be sedentary in leisure time [17]. It is estimated that if all people were to reach acceptable levels of physical activity, main chronic diseases would diminish between 5–10% while live expectancy would increase [16]. We could also add the health cost of sedentarism—estimated at 53.8 billion dollars—as well as the 13.7 billion lost in productivity and the 13.4 million lost in disability-adjusted life years [18]. At the local level, in accordance with the Health Economic Assessment Tool (HEAT), it was estimated that 32.1 million of euros could be saved in one year if 5% of commuting by motor vehicles was done walking or cycling [19]. Environments with proper infrastructures for walking favor people to move more on foot, thus increasing the amount of physical activity per person, relieving traffic congestion and reducing both noise and air pollution [20,21,22,23,24], and consequently, leading to a decrease in morbidity attributable to contamination [25].

On the other hand, walking depends not only on the person, but also on the access to basic physical and social environment resources where people live [26,27]. Certain features of the urban environment could stimulate or discourage the habit of walking [26,28]. In this way, the concept of walkability arises, which is defined as the degree in which the urban environment promotes walking [29]. Features influencing walkability are security, esthetic, traffic, design and connectivity between streets, building density, mixed-use streets, possibility of walking from one place to another considering accessibility and the provision as well as the distance to public facilities and outdoor spaces, among others. These aspects are essential to establish not only measures of promotion of daily physical activity [30,31,32,33,34], but also other recreational activities in public spaces [35]. Moreover, people living close to a place that favors the practice of physical activity are more likely to do so [36].

Walkability indices usually correlate with the amount of physical activity performed, but not in all environments [37]. A meta-analysis showed that safety, walkability and aesthetics influenced physical activity in adults. However, when the different variables that defined each of these components were studied separately, the resulting evidence was not strong enough due to the heterogeneity of instruments that measure the amount of physical activity, the methodological differences and the great variability of infrastructures and environments. For this reason, it is necessary to continue studying the underlying mechanisms associated between walkability and environment [38]. In this sense, several procedures have been described to evaluate the environment with respect to walkability: (a) questionnaires of perception, with a high level of subjectivity from the observer, such as Neighborhood Environment Walkability Survey (NEWS), neighborhood quality index and perceived walking environment questionnaires; (b) observation and measurements through audits, such as Systematic Pedestrian and Cycling Environmental Scan (SPACES), Senior Walking Environmental Assessment Tool (SWEAT), Irvine-Minnesota (I-M) and Pedestrain Environment Data Scan (PEDS) questionnaires; and (c) objective measurements with geographic information systems (GIS) assessing population density, land-use mix, access to recreational facilities, street pattern, sidewalk coverage, vehicular traffic, other (e.g., building design, public transit, slope, greenness/vegetation) [39,40]. However, for the present time, these have not been translated to Spanish nor validated in the Spanish field. Moreover, they are more rigid and difficult to answer as they need specific equipment to measure the width of pavements, the slope or the noise pollution. Furthermore, most Spanish cities have mediaeval city centers, which were built before the existence of motor vehicles. Consequently, they present an irregular urban morphology, narrow streets and no pavements in general, in contrast with the straight and large streets of the modern and planned cities, on which the other walkability questionnaires were validated. In the last year, GIS were helpful for in vivo audits, saving time in measuring the most objective aspects. Gullón [41] observed a good correlation between items evaluated with GIS and trough audits, especially when permeability of infrastructure, traffic safety and destinations were evaluated. Nonetheless, the esthetic aspects were better evaluated through direct observation [42]. GIS are limited, for they are not sensitive to the changes that occur after the digitalization of the images and may show low quality characteristics.

The World Health Organization points out the importance of the people’s involvement in health-promoting community activities [43,44]. Community activities are effective for health promotion [45], but their development in Spain varies greatly in terms of ways of implementation, institutional support and involvement of the community [46]. Hence, the community’s health should be developed from the territorial area closest to the citizen—the local level—as it can take advantage of every sector involved in the health of the community [44,47]. Consequently, the Ministry of Health of the Balearic Islands developed the healthy routes program keeping community participation in mind. This program aimed to promote physical activity through walks in the neighborhood, close to the health center, ensuring the safety of the population. The different routes were designed by a committee specifically constituted for this matter, which was comprised of visitors from the health center; neighbors, associations and charities from the neighborhood; urban planning specialists from the city council; and health professionals employees from the General Direction of Public Health as well as from the health services of the Balearic Islands. The objective of the study was to validate the urban walkability perception questionnaire (UWPQ) in the Balearic Islands, which was created within the framework of the “healthy routes” community participation program. The purpose of this instrument was to determine the security and adequacy of the characteristics of the urban environment to define routes that promote walking among the population.

## 2. Materials and Methods

### 2.1. Analytical Method

A panel of independent experts in geography, architecture, engineer, public health and the community citizens created the UWPQ in Spanish (Appendix A) based on the NEWS questionnaire—due to its easy understanding—and on other previous validated questionnaires dealing with perception of the environment [40]. The UWPQ is structured in three blocks with 22 items in total, plus an additional Item 23 regarding the participant’s general opinion on each section of the evaluated route. The three blocks, with a global vision of the concept of walkability, are as follows:Pedestrian facilities (Items 1–11). This block involves the presence of pedestrian-only streets, clearly marked pavements and streets for both pedestrians and vehicles; their condition and width; as well as the presence of obstacles, pedestrian-dropped kerbs, garage entrances and exits, slopes and stairs, well-signalized crossings and bicycle lanes.Infrastructures of the environment (Items 12–19). This block refers not only to the existence of benches, lighting, trees and other elements that may favor walking, but also to the state of cleanliness, noise, traffic density, parks and other public open spaces.Perception of safety (Items 20–22). This block aims to evaluate how safe the participants perceive the route to be when crossing the street and during the daytime and nighttime.

In order to force the respondents to position themselves towards a disagreement or agreement value and avoid intermediate answers, a 4-category Likert scale was used with the following score: totally agree—1 point; strongly agree—0.75 point; disagree—0.25 point and completely disagree—0 points. This score applied for all items, except for Items 8, 9, 16 and 17, which were scored inversely. In addition, Item 18 referred to the existence of assets related to outdoor physical activities that could stimulate walking (such as sport areas, squares, parks and beaches). Each of these assets was scored as Yes/No and subsequently transformed into 1/0, which resulted in the total number of assets. Finally, a total score was obtained for each section of the route.

For the health routes committee to establish the routes, these had to be 2400-m-long at least [48], be circular if possible and have only one pavement in one direction. Then, the route was divided into homogeneous sections, that is, parts of a street between two intersections. The committee was trained on how to rate the different aspects of the UWPQ: accessibility, security and maintenance of infrastructure.

To make the validation process more efficient, we performed a parallel audit with the Street View viewer [48,49] to locate those sections that a priori could accomplish high levels of walkability (large pavements, low traffic density, signalized crossings, lack of obstacles, etc.).

The UWPQ was piloted in a 3239 m route of the Escola Graduada Health Center, located in Palma’s old town, so as to analyze its reliability, interobserver agreement, incidences and doubts. The results were discussed with the panel of experts to define the definitive questionnaire for the validation process. The manual was also reviewed and improved.

Finally, the validation was performed in 12 routes by adults with no illnesses preventing them from walking at 5 km per hour. They were from 12 urban neighborhoods associated with 12 health centers of the main cities in the Balearic Islands. Ten routes were evaluated in Palma, on the island of Mallorca, which has a population of 409,661 inhabitants and a significant urban variability, with very old neighborhoods from medieval times, rehabilitated neighborhoods and other more modern ones. Another route was validated in Ciutadella, the capital of the island of Menorca, with a population of 29,223 inhabitants. Finally, another route belongs to Ibiza, on the island of Ibiza, with a population of 49,727 inhabitants. These last two routes combine two urban areas: a more modern one of the seaport and another one that crosses through the central areas of the cities, which are older. For more details, you can consult https://e-alvac.caib.es/es/rutas-portada.html.

A total of 190 people are recruited from a process of community participation to audit the routes, 18 of which do not get to answer the questionnaires with the minimum necessary information. All of the subjects consented their participation in the study.

### 2.2. Statistical Analyses

Based on other walkability validation studies [50,51,52,53], a descriptive analysis (mean, median, mode and standard deviation) was performed with the score of each item. Correlation grade [54] between items was calculated with the nonparametric Spearman’s correlation coefficient.

Reliability was analyzed using Cronbach’s alpha internal consistency coefficient [55] and the Homogeneity Index, which measured the correlation between the 22 items and the total score. The corrected Homogeneity Index was also calculated [56].

Internal validity was measured with the Goodman and Kruskal’s gamma correlation coefficient (γ) [57] between Items 1–22 of the questionnaire and with Item 23, the overall assessment of the section. The purpose of the latter was to validate the relationship between the score of the 22 items and the global one.

SPSS Statistics version 17 (IBM, Armonk, NY, USA) was used. Statistical significance was established at *p* < 0.05.

## 3. Results

A total of 172 people (67.4% women and 32.6% men) participated, who audited 566 sections in a total of 12 routes. Descriptive analysis by item is shown in Table 1.

As for the pavement, 84% of the sections allowed walking comfortably and 78% of them did not have many obstacles such as cars on the pavement, street furniture, light poles, streetlamps or garbage containers. However, only 19.5% of the sections were pedestrian-only streets. Nevertheless, 88.6% of the routes had clearly marked pavements, 81% had pedestrian dropped kerbs, 86% had no excessive slopes or stairs, and 83.5% had pedestrian crossings properly signalized. On the other hand, one out of three participants (33%) claimed to feel insecure while walking on pavements with a bicycle lane aside, and 40% pointed out at the lack of benches to rest while 45% noted the poor shade. Lastly, 66% of the participants agreed that, globally, the pavements, streets and buildings were clean.

Regarding the services identified, 73.3% of sections had bars, cafes or restaurants, 60.8% had local shops and 57.6% had a bus stop nearby. In addition, there were supermarkets in 47.9% of the sections; local services, such as banks or post offices, in 43.6% of them; and outdoor recreation areas in 34.6% of them. Finally, 30.7% of the sections had a cultural building—like a museum—28.3% had recreational centers and 19.1% had sports centers.

Regarding Item 23 on the general opinion of the section, 37.3% of the participants found the route was totally nice while 43% of them deemed it quite nice; on the other hand, 15.6% of them considered it was not much nice and 4.5% of them concluded it was not nice at all.

### 3.1. Degree of Correlation

The degree of correlation between the items is shown in Appendix B. The moderate linear correlations with a significance level <0.01 were the following:

From the pedestrian facilities block, Item 6—*Is it possible to walk without obstacles?* correlated with Item 5—*Is it large enough to walk comfortably?* (rs = 0.600), while Item 7—*Do the pavements have adequate pedestrian dropped kerbs to comfortably cross from one corner to the other?* correlated with Item 4—*Are the maintenance and conservation conditions of the pavement suitable for walking?* (rs = 0.512) and with Item 5—*Is it large enough to walk comfortably?* (rs = 0.518).

From the Infrastructures of the environment block, Item 17—*Is it a section with high traffic density? correlated with Item 16—Is there too much noise that makes walking uncomfortable?* (rs = 0.536).

From the Perception of safety block, Item 20—*Is it possible to cross the streets safely?* correlated with Item 5—*Is it large enough to walk comfortably?* (rs = 0.537) and with Item 10—*Are the pedestrian crossings well signalized?* (rs = 0.536), while Item 22—*The section is safe to walk at night* correlated with Item 21 *The section is safe to walk during the day* (rs = 0.579).

Lastly, Item—23 *In general, do you think this section of the route is nice for walking?* correlated with Item 6—*Is it possible to walk without obstacles?* (rs = 0.529).

As for the most remarkable inverse linear correlations, Item 8—*Is there an excessive presence of entries and exits for vehicles posing a threat?* inversely correlated with Item 5—*Is it large enough to walk comfortably?* (rs = −0.363) and with Item 6—*Is it possible to walk without obstacles?* (rs = −0.401); while Item 10—*Are the pedestrian crossings well signalized?* did so with Item 8—*Is there an excessive presence of entries and exits for vehicles posing a threat?* (rs = −0.377) and with Item 9—*Does the section have an excessive slope or steps that make walking difficult?* (rs = −0.323). All correlations were significant (*p* < 0.01).

### 3.2. Reliability

Internal consistency was 0.7 in all blocks according to the Cronbach’s alpha test (*p* < 0.01). Item 1—*Is it a pedestrian-only street?* and Item 9—*Does the section have an excessive slope or steps that make walking difficult?*—both from the pedestrian facilities block—showed a corrected item-total correlation lowest of 0.022 and 0.096, respectively.

The index of homogeneity and the corrected index of homogeneity are shown in Table 2. All the items, except for Item 18—*Number of elements of the section that stimulate walking*, obtained values higher than 0.15 (*p* < 0.01).

In order to improve the internal consistency, Items 1 and 9—from the pedestrian facilities block—were excluded while Items 8, 16, 17 and 18 were reformulated. On the other hand, Item 10—*Are the pedestrian crossings well signalized?* was moved to the Perception of safety block. Thus, by redefining the questionnaire, the internal consistency of the pedestrian facilities block increased up to 14%, α = 0.8. The final questionnaire with 26 items is shown in Appendix C.

### 3.3. Internal Validity

Goodman and Kruskal–gamma correlation coefficient (γ) is shown in Table 3. All correlations between Item 23 and the rest of the items were significant. The items from the Perception of safety and pedestrian facilities blocks that collected the easiness to walk (Items 4–7, 10 and 11) were the ones that most affected the final assessment of the section. The items regarding the pedestrian-only pavements, clearly marked pavements, noise, traffic density and parks condition (Items 1, 2, 16, 17 and 19, respectively) obtained the lowest coefficients.

## 4. Discussion

The purpose of this study was to validate an instrument designed to evaluate the adequacy of the urban environment and infrastructures that could enable the design of new routes for encouraging the adult population of the Balearic Islands to perform more physical activity by walking. The selected sections obtained mostly a very good score, probably because the participants chose the most favorable streets for walking, as they lived in the neighborhood where the validation took place and they were familiar with it. Therefore, the community participation process prior to the validation was an efficient starting point, as neither time nor resources were wasted in planning the routes with people not familiar with the environment. As in Chen’s study [58], the characteristics of the chosen routes prove this assumption, since most of the pavements and pedestrian streets had a limited number of obstacles, were large enough to walk comfortably and had adequate pedestrian dropped kerbs, while the crossings were well signalized as well.

One third of the participants claimed that they felt unsafe when the section was beside a bicycle lane. It is likely that this result was due to the age of the participants and to the fact that they felt unprotected when cyclists passed by very closely because of their lack of visual acuity and balance. As a matter of fact, Palma has experienced a huge increase of bicycle lanes over the last years. However, most of them are located on the pavements instead of having their own road and, consequently, they were perceived as a poor structure of safety. This was also found by Sawchuk [59]. As most of the users of these routes are aged people, this aspect should be considered in the design of new routes.

In most of the routes, it was highly probable to find services and shop infrastructures. However, the presence of outdoor spaces as squares or leisure facilities was less frequent. Even though, few sections were assessed as not much nice or not nice at all.

The correlations between items were significant in the pedestrian facilities and Perception of safety blocks, whereas in the Infrastructures of environment block, the correlations were weak and non-significant. For this reason, we considered that the items that assessed walkability directly—as the pavement and immediate environment conditions—were the ones that really promote walking, and therefore, these were the most related between them.

Regarding to the UWPQ’s reliability, the obtained results confirmed that the distribution in three blocks for the 22 items presented a good internal consistency. Therefore, it was normal that the items that assessed if the pavement was large enough and had adequate pedestrian dropped kerbs and no obstacles were highly and positive correlated. It was also reasonable that the items referring to the presence of enough benches, light and trees were included in the Infrastructures of environment block that promotes walking [59,60].

Nevertheless, it would be necessary to exclude some items that were irrelevant or were repeated in other items. Specifically, Item 20—*Is it possible to cross the streets safely?*—from the Perception of safety block—and Item 10—*Are the pedestrian crossings well signalized?*—from the pedestrian facilities block—showed a high correlation; so, these two blocks were redefined, and Item 10 was introduced in the Perception of safety block.

Furthermore, neither Item 1—*Is it a pedestrian-only street?* nor Item 9—*Does the section have an excessive slope or steps that make walking difficult?*—both from the pedestrian facilities block—were well correlated with the rest of items of that block nor had a significant effect with Item 23 on the general opinion of the section. When these were removed, the internal consistency of the block improved, so they were finally excluded from the questionnaire. We attribute this to the fact that the participants considered more important that the pavements were large enough, for the sole use of pedestrians and had no obstacles, since sometimes the streets were crowded or occupied with furniture from bars and restaurants, making it difficult to walk comfortably. Moreover, we must consider that, before the assessment, the routes were studied in detail with the geographic viewer Street View to evaluate certain characteristics and ease the in situ validation process [41,42]. As in Mooney [49], the participants chose sections without slopes or steps to avoid physical overexertion or to prevent injuries.

As for Item 18—*Number of elements of the section that stimulate walking*, it provided a numeric answer option with a list of elements such as sports centers, gyms, swimming pools, supermarkets, bars, restaurants, etc. After the validation, we decided that this item had to stay in the questionnaire, but only after reviewing the wording and modifying the answers to a Likert scale since, according to Riera-Sampol [61], this contributes to the process of health assets mapping from health centers.

We identified certain items that influenced more than others the final perception on how appealing the section was for walking. These items were: a large enough pavement, the absence of obstacles and well signalized and safe pedestrian crossings. Thus, the participants may prioritize their safety and comfort over the esthetics.

The UWPQ is in consonance with other instruments that were validated in the Anglo-Saxon field, such as SPACES [62], PEDS [40], I-M [63] and NEWS [64].

### Limitations

The analysis has certain limitations derived from measuring the items with a four-value Likert scale. On one hand, the statistical analysis was limited to the tests for ordinal and nonparametric data and, on the other hand, the weight of each item in the questionnaire was identical. Moreover, we cannot avoid the influence of the participants’ opinion in the assessment of the items. An excess of subjectivity could reduce the internal consistency of the instrument. However, the experts group tried to avoid complicated questions and technical terms.

## 5. Conclusions

The obtained results showed that the UWPQ is a suitable instrument to assess the degree of adequacy of the urban environment for walking. The methodology used to create and validate the questionnaire, with a panel of experts and a community participative process, fits well.

The UWPQ could contribute to create environments favorable for the health and wellbeing of the population as well as to improve the coordination with other public policies, such as the Department of Housing and Urban Planning, the decisions of which may affect people’s health. For this reason, walkability should be considered as a criterion for urban planning projects as well as for sustainable development. This could also help to reach the Sustainable Development Goals [65].

Identifying which characteristics of the physical environment stand out to determine if a route is safe and healthy could promote the habit of walking at least half an hour daily among the population, thus complying with the World Health Organization recommendations.

## Figures and Tables

**Table 1 ijerph-17-06631-t001:** Description of items of the urban walkability perception questionnaire (UWPQ).

	% Answers	Average	Median	Mode	Standard Deviation
**A. PEDESTRIAN FACILITIES**					
1. Is it a pedestrian-only street?	93.29	2.79	3.00	4.00	1.19
2. Is there a clearly marked pavement for walking?	92.76	1.50	1.00	1.00	0.81
3. If the street is for both pedestrians and vehicles and has no pavement, is it safe for walking? (If there is pavement, mark “It does not apply”)	30.74	1.95	2.00	2.00	0.90
4. Are the maintenance and conservation conditions of the pavement suitable for walking?	93.29	1.97	2.00	2.00	0.80
5. Is it large enough to walk comfortably?	91.87	1.73	1.50	1.00	0.86
6. Is it possible to walk without obstacles?	92.76	1.90	2.00	2.00	0.87
7. Do the pavements have adequate pedestrian dropped kerbs to comfortably cross from one corner to the other? (If there is no pavement, mark “It does not apply”)	88.69	1.80	2.00	1.00	0.84
8. Is there an excessive presence of entries and exits for vehicles posing a threat?	92.05	3.01	3.00	3.00	0.89
9. Does the section have an excessive slope or steps that make walking difficult?	93.46	3.35	4.00	4.00	0.88
10. Are the pedestrian crossings well signalized?	91.87	1.75	2.00	1.00	0.84
11. If the pavement shares space with a bicycle lane, is it safe to walk in this section? (If there is no bike lane, mark “It does not apply”)	47.17	2.22	200	2.00	1.01
**B. INFRASTRUCTURES OF THE ENVIRONMENT**					
12. Are there enough benches for resting?	91.70	2.31	2.00	2.00	1.10
13. Does the section have enough light at night?	82.69	1.80	2.00	2.00	0.75
14. Are there enough trees providing shade and thus allowing to walk comfortably during the hours of sun?	93.46	2.46	2.00	2.00	0.99
15. In general, are the pavements, streets and buildings of the section clean?	93.29	2.23	2.00	2.00	0.80
16. Is there too much noise that makes walking uncomfortable?	93.11	2.77	3.00	3.00	0.74
17. Is it a section with high traffic density?	92.76	2.54	3.00	3.00	0.80
18. Number of elements of the section that stimulate to walk (yes/no question) Sports centers, gyms, swimming pools, etc. Outdoor recreational areas such as parks, beaches, etc. Recreation centers, etc. Local businesses (groceries, pastry, hairdresser, pharmacy, etc.) Supermarket Local services (bank, post, etc.) Bars, cafes, restaurants, etc. Bus stop Attractive buildings, museums, heritage elements, churches, cultural centers, etc.	100.00	4.27	5.00	6.00	2.58
19. Are the parks, gardens and other public open spaces in good condition? (If there are none, mark “It does not apply”)	75.27	2.08	2.00	2.00	0.82
**C. PERCEPTION OF SAFETY**					
20. It is possible to cross the streets safely.	88.69	1.80	2.00	2.00	0.72
21. The section is safe to walk during the day.	88.52	1.77	2.00	2.00	0.74
22. The section is safe to walk at night.	76.15	2.13	2.00	2.00	0.85

**Table 2 ijerph-17-06631-t002:** Homogeneity index and corrected index of homogeneity for the items of the UWPQ.

	Homogeneity Index	Corrected Index of Homogeneity
**A. PEDESTRIAN FACILITIES**		
1. Is it a pedestrian-only street?	0.367 **	0.255 **
2. Is there a clearly marked pavement for walking?	0.382 **	0.306 **
3. If the street is for both pedestrians and vehicles and has no pavement, is it safe for walking? (If there is pavement, mark “It does not apply”)	0.49 **	0.424 **
4. Are the maintenance and conservation conditions of the pavement suitable for walking?	0.498 **	0.435 **
5. Is it large enough to walk comfortably?	0.652 **	0.595 **
6. Is it possible to walk without obstacles?	0.656 **	0.599 **
7. Do the pavements have adequate pedestrian dropped kerbs to comfortably cross from one corner to the other? (If there is no pavement, mark “It does not apply”)	0.556 **	0.489 **
8. Is there an excessive presence of entries and exits for vehicles posing a threat?	0.482 **	0.403 **
9. Does the section have an excessive slope or steps that make walking difficult?	0.442 **	0.370 **
10. Are the pedestrian crossings well signalized?	0.563**	0.496**
11. If the pavement shares space with a bicycle lane, is it safe to walk in this section? (If there is no bike lane, mark “It does not apply”)	0.544 **	0.461 **
**B. INFRASTRUCTURES OF THE ENVIRONMENT**		
12. Are there enough benches for resting?	0.526 **	0.432 **
13. Does the section have enough light at night?	0.478 **	0.413 **
14. Are there enough trees providing shade and thus allowing to walk comfortably during the hours of sun?	0.386 **	0.293 **
15. In general, are the pavements, streets and buildings of the section clean?	0.543 **	0.483 **
16. Is there too much noise that makes walking uncomfortable?	0.296 **	0.229 **
17. Is it a section with high traffic density?	0.275 **	0.200 **
18. Number of elements of the section that stimulate to walk (yes/no question) Sports centers, gyms, swimming pools, etc. Outdoor recreational areas such as parks, beaches, etc. Recreation centers, etc. Local businesses (groceries, pastry, hairdresser, pharmacy, etc.) Supermarket Local services (bank, post, etc.) Bars, cafes, restaurants, etc. Bus stop Attractive buildings, museums, heritage elements, churches, cultural centers, etc.	−0.049	0.031
19. Are the parks, gardens and other public open spaces in good condition? (If there are none, mark “It does not apply”)	0.524 **	0.465 **
**C. PERCEPTION OF SAFETY**		
20. Is it possible to cross the streets safely?	0.609 **	0.557 **
21. The section is safe to walk during the day	0.547 **	0.495 **
22. The section is safe to walk at night	0.520 **	0.454 **

** *p* < 0.01.

**Table 3 ijerph-17-06631-t003:** Coefficients of Goodman and Kruskal’s gamma correlation between the items of the UWPQ and global assessment of the section (Item 23).

	Value	Standard Deviation	T	Signification
**A. PEDESTRIAN FACILITIES**				
1. Is it a pedestrian-only street?	0.243	0.059	4.102	0.000
2. Is there a clearly marked pavement for walking?	0.66	0.067	3.788	0.000
3. If the street is for both pedestrians and vehicles and has no pavement, is it safe for walking? (If there is pavement, mark “It does not apply”)	0.302	0.107	2.753	0.000
4. Are the maintenance and conservation conditions of the pavement suitable for walking?	0.529	0.051	9.272	0.000
5. Is it large enough to walk comfortably?	0.644	0.043	11.938	0.000
6. Is it possible to walk without obstacles?	0.677	0.040	13.691	0.000
7. Do the pavements have adequate pedestrian dropped kerbs to comfortably cross from one corner to the other? (If there is no pavement, mark “It does not apply”)	0.497	0.050	9.081	0.000
8. Is there an excessive presence of entries and exits for vehicles posing a threat?	0.489	0.049	9.046	0.000
9. Does the section have an excessive slope or steps that make walking difficult?	0.431	0.055	7.013	0.000
10. Are the pedestrian crossings well signalized?	0.605	0.044	11.668	0.000
11. If the pavement shares space with a bicycle lane, is it safe to walk in this section? (If there is no bike lane, mark “It does not apply”)	0.561	0.059	8.228	0.000
**B. INFRASTRUCTURES OF THE ENVIRONMENT**				
12. Are there enough benches for resting?	0.510	0.046	10.099	0.000
13. Does the section have enough light at night?	0.485	0.054	8.210	0.000
14. Are there enough trees providing shade and thus allowing to walk comfortably during the hours of sun?	0.323	0.055	5.642	0.000
15. In general, are the pavements, streets and buildings of the section clean?	0.555	0.045	10.763	0.000
16. Is there too much noise that makes walking uncomfortable?	0.298	0.060	4.877	0.000
17. Is it a section with high traffic density?	0.214	0.060	3.498	0.000
18. Number of elements of the section that stimulate to walk (yes/no question) Sports centers, gyms, swimming pools, etc. Outdoor recreational areas such as parks, beaches, etc. Recreation centers, etc. Local businesses (groceries, pastry, hairdresser, pharmacy, etc.) Supermarket Local services (bank, post, etc.) Bars, cafes, restaurants, etc. Bus stop Attractive buildings, museums, heritage elements, churches, cultural centers, etc.	0.527	0.055	8.598	0.000
19. Are the parks, gardens and other public open spaces in good condition? (If there are none, mark “It does not apply”)	0.208	0.057	3.588	0.000
**C. PERCEPTION OF SAFETY**				
20. Is it possible to cross the streets safely?	0.652	0.046	11.517	0.000
21. The section is safe to walk during the day	0.596	0.048	10.761	0.000
22. The section is safe to walk at night	0.492	0.055	8.263	0.000

*p* < 0.01.

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
