# Peer review of "Validation of the Urban Walkability Perception Questionnaire (UWPQ) in the Balearic Islands"

_ijerph, 2020, doi:10.3390/ijerph17186631_

Round 1

Reviewer 1 Report

The paper takes the walkability of the built environment as the research objective, and tests the validity of the Urban Walkability Perception Questionnaire (UWPQ). The paper conforms to the current healthy city development and has certain value. But the following problems still exist:

  1. The introduction part has too many paragraphs and the logic is not clear. It is suggested that the part of introduction has four paragraphs, including research background, research significance, literature review and research objective.

  1. Literature review cannot support research topic. The topic of the paper is the validation of walkability assessment, but the literature review part only mentions the measurement and limitations of built environment characteristics. It is suggested to add the content of the existing walkability evaluation methods and their application, and discussing the selection of UWPQ as the basis for evaluating the walkability of Balearic Islands.

  1. In Materials and Methods, it is suggested to add the introduction of study site, especially the built environment characteristics for walking. Based on the conclusions of the literature review, the reason to choose UWPQ as a method to assess the walkability of Balearic Islands needs to be discussed in more detail.

  1. It is suggested to use Research/Analytical Method as the title of 2.1. In this part, it is suggested to add the sampling procedures, the number of questionnaires issued and the number of valid questionnaires and other data collection.

  1. In the second paragraph of 4. Discussion,the paper mentions that One third of the participants claimed that they felt unsafe when the section was beside a bicycle lane. It is likely that this result was due to the age of the participants and to the fact that they felt unprotected when cyclists passed by very closely because of their lack of visual acuity and balance. One of the explains is that socioeconomic attributes at the individual level such as age, gender, and education level will affect residents’ perception of the safety in built environment. It is suggested to use quantitative models to analyze the causal relationship between the items in UWPQ and the walking behavior, with controlling the individual socioeconomic factors.

  1. This paper needs further proofreading. The text contains many linguistic errors (such as singular and plural problems, lack of subject or predicate, use of term, etc.), which may sometimes make it difficult to understand. After modifying the content of the paper, it may be helpful to have a professional English editor to polish the language of the paper.

Author Response

Please see the attachment for our responses. We have also sent the manuscript to our editor as this is where we have made corrections to improve it.

Reviewer 2 Report

This is surely an interesting piece of research but cannot be published at its current state. My comments are shown below.

First, I do not see much "healthy" factor in the questionnaire, represented by the questions shown in Appendix A. The questions are more about physical conditions and safety and do not directly discuss health issues. Of course, promoting health is a key benefit from walking but the questions seem to be looking at other issues.

Second, my question is about the usefulness of validation. Why does the questionnaire need to be validated? Also, methods used for validation must be justified. I don't think the statistical tools adopted in this study are not the only validation measures.

Third, I question to what extent findings of this study can be applied elsewhere. I understand that this work comes from the Balearic Islands, but there must be some valuable implications or lessons.

Fourth, why was a 4-point Likert scale used? It is okay but not usual. Justification should be provided. 

Several minor issues:

It would be better if the questions are presented the main context of this paper not in the appendix section. 

The literature review section should be revised. It can also be separated from Introduction.

Author Response

(The authors gave the same response as above.)

Reviewer 3 Report

This article presents a questionnaire-based approach to evaluate walkability in the Balearic Islands (Spain). The authors focus on the validity of such and approach and the benefits/limitations of their questionnaire while also discussing the results from their on-field analysis.

Although the article is quite short and results/methods simple and generic it does what it is expected and what is promised from the reading of the title. So from this point of view I don’t see any reason to reject this article and although I believe it may not become a milestone in the literature it nonetheless nicely contribute to the field of walkability and represents a good example for an area not heavily urbanized (usually walkability is assesses in large cities or modern areas as the authors pointed out, with semi-rural areas usually less investigated).

I just have a few comments which are listed as follows:

  1. Abstract: is it necessary to provide the results of the internal consistency and gamma correlation here? I don’t know if readers would need such information while checking the abstract. I think background, methods and conclusions are the most important aspects in the abstract, with less relevant numerical results only reported if are one of the main outcomes.
  2. Line 112: I am not an expert in questionnaire, although I do have some experience and generally understand your approach. But I believe it is arguable if you could set a positive score for disagreeing. I personally would have set a negative score for disagree and completely disagree to reflect the fact that opinion is the opposite to people agreeing on it. However, again, I am not an expert so I can probably not judge on this. I think a rationale or a reference is needed to justify your scoring approach. I know that by changing it all results will change, so I am not asking to change this aspect, but since all results are based on this, it is important that readers understand why you choose those values.
  3. In the questionnaire: “It doesn’t proceed”: I understand what you mean, but probably there is a better translation for this. Maybe you want to say “It doesn’t apply”.
  4. Line 161: very minor comment (mostly personal). I don’t know if museums are actually touristic places. Maybe in the Balearic Islands they take this function, but generally museums are intended a mean to provide culture and education to the local population and only few and very famous ones are for tourists (probably here the image as touristic places).
  5. Section 3.1 and following: when you provide the questions in the text you should either add a “ before and after the question or employ the italic approach used later. If text style is the same as the text it is difficult to distinguish a question taken from the questionnaire from the text of the manuscript.
  6. Lines 200-204: this part is quite confusing. You mention that the questionnaire was improved and some discussion on this aspect is also given later on. So which questionnaire are you using in this study? On which version of the questionnaire are the results based on? In my understanding this paper is based on the old version and based on its results you modified the questionnaire for future use. But since you say that internal consistency was improved it means you also have data from the new questionnaire. The differences between the new one and the old one and the source of the data should be more clear.

Author Response

(The authors gave the same response as above.)

Round 2

Reviewer 2 Report

It is difficult to identify what has been revised based on my previous comments. As a reviewer, I sincerely ask the authors to carefully revise the manuscript by incorporating my comments and not just rebut via the response file.

Author Response

Response to Reviewer 2 Comments

Point 1. I do not see much "healthy" factor in the questionnaire, represented by the questions shown in Appendix A. The questions are more about physical conditions and safety and do not directly discuss health issues. Of course, promoting health is a key benefit from walking but the questions seem to be looking at other issues.

The objective is to validate a questionnaire that allows auditing the walkability of the environment. It is not a questionnaire that evaluates direct aspects of health.

When public health technicians must take actions to improve the health of the population, we often have to do it indirectly, as in this case. For us, the ultimate goal is to improve the health of the population, in this case by improving the neighbourhood environment so that people can walk comfortably, without obstacles or other impediments that pose a danger. For an environment to be healthy, it first must be safe without predisposing to accidents, otherwise it makes no sense. The set of questions, almost all referring to aspects of urban planning and infrastructures (presence of sports centers, gyms, swimming pools, recreational areas, cultural buildings ...) refer to the health assets that are important when obtaining a final judgment so that the person feels stimulated to go walking, and indirectly can do daily physical activity to improve their health.

We have tried to explain it in the Introduction section of the manuscript from line 32 to 56; from line 90 to 91 and from  98 to 100 (in “simple revision mode”). Changes appear by “control changes” and also highlighted in yellow.

Point 2. My question is about the usefulness of validation. Why does the questionnaire need to be validated? Also, methods used for validation must be justified. I don't think the statistical tools adopted in this study are not the only validation measures.

We believe that the validation of the questionnaire is essential because through a systematic and scientific process we submit an instrument to analysis to verify its validity and reliability. Thus, we are certain that we are using a first and only questionnaire that assesses walkability and can be reproduced in a Spanish context.

Secondly, the methods used are those typical of the questionnaire validation processes, such as the degree of correlation, reliability and internal validity. For this type of questionnaire, there would be no better processes.

In the point 2.2. Statistical analyses of the manuscript, we have inserted eight bibliographic references (references 50-57) that support the use of our psychometric process. They are highlighted in yellow.

Point 3. I question to what extent findings of this study can be applied elsewhere. I understand that this work comes from the Balearic Islands, but there must be some valuable implications or lessons.

According to the bibliographic review, to date there is no validated questionnaire in a Spanish environment. The purpose of validating the UWPQ is to be able to have an instrument agreed upon with various sectors of the community to unify which are the best criteria to evaluate the environment to favour walkability. From the General Directorate of Public Health of the Balearic Islands, they want to extend the promotion of physical activity for the entire community and, having a valid instrument, means that all health centers can ensure that they have a route that effectively measure how favourable your environment is for practicing physical activity.

We have explained it in the Introduction section of the manuscript from line 71 to 77 (in “simple revision mode”). Changes appear by “control changes” and also highlighted in yellow.    

Point 4. Why was a 4-point Likert scale used? It is okay but not usual. Justification should be provided. 

We have used a 4-point scale to avoid the central tendencies of the scales with 5, 7 or 9 points. According to our statistician, it is currently more reliable to force respondents to answer towards a state of disagreement or agreement and thus avoid scores with mean values. 

We have explained it in the Analytical method section of the manuscript from line 117 to 120 (in “simple revision mode”). Changes appear by “control changes” and also highlighted in yellow.

Several minor issues:

It would be better if the questions are presented the main context of this paper not in the appendix section. 

We have done it this way so as not to fragment the reading of the text with the tables too much since they are very long. That is why we have decided to combine some tables in the text and the appendix to give more harmony and fluency to the reading of the text.

The literature review section should be revised. It can also be separated from Introduction.

We agree. We have improved the text and have made changes it in the manuscript. Changes appear by “control changes”.